# Current Status and Future Perspectives of Nuclear Medicine in Prostate Cancer from Imaging to Therapy: A Comprehensive Review

**DOI:** 10.3390/biomedicines13051132

**Published:** 2025-05-07

**Authors:** Joohee Lee, Taejin Kim

**Affiliations:** 1CHA Ilsan Medical Center, Department of Nuclear Medicine, CHA University College of Medicine, Ilsan 10414, Gyeonggi-do, Republic of Korea; ljhnm04@chamc.co.kr; 2CHA Ilsan Medical Center, Department of Urology, CHA University College of Medicine, Ilsan 10414, Gyeonggi-do, Republic of Korea

**Keywords:** nuclear medicine, prostate cancer, SPECT, PET/CT, theranostics

## Abstract

Nuclear medicine has emerged as a critical modality in the diagnostic and therapeutic management of urological malignancies, particularly prostate cancer. Advances in single-photon emission computed tomography/computed tomography (CT) and positron emission tomography/CT (PET/CT) have enhanced tumor assessment across staging, treatment response, and recurrence settings. Molecular imaging, which offers insights beyond traditional anatomical imaging, is increasingly integral in specific clinical scenarios. Theranostic nuclear medicine, which combines diagnostic imaging with targeted therapy, has become a well-established treatment option, particularly for patients with metastatic castration-resistant prostate cancer (mCRPC). The development of the prostate-specific membrane antigen (PSMA) radioligands has revolutionized clinical management by enabling precise disease staging and delivering effective radioligand therapy (RLT). Ongoing research aims to refine the role of PSMA PET imaging in staging and treatment monitoring, while optimizing PSMA-targeted RLT for broader clinical use. Given that prostate cancer remains highly prevalent, the anticipated increase in the demand for RLT presents both challenges and opportunities for nuclear medicine services globally. Theranostic approaches exemplify personalized medicine by enabling the tailoring of treatments to individual tumor biology, thereby improving survival outcomes and maintaining patients’ quality of life with minimal toxicity. Although the current focus is on advanced disease, future research holds promise for expanding these strategies to earlier stages, potentially enhancing curative prospects. This evolving field not only signifies a paradigm shift in the care of prostate cancer patients but also underscores the growing importance of nuclear medicine in delivering precision oncology.

## 1. Introduction

Prostate cancer (PCa) is among the most prevalent malignancies in the male population and the second leading cause of cancer-related mortality worldwide [1,2,3]. Its biological behavior is highly diverse, ranging from indolent, slow-growing tumors to highly aggressive disease [4]. Consequently, risk stratification has become a fundamental component of personalized treatment planning, enabling the development of tailored therapeutic approaches based on the patient’s prognosis [5,6].

The diagnostic evaluation of PCa traditionally includes serum prostate-specific antigen (PSA) testing and digital rectal examination (DRE), followed by transrectal ultrasound (TRUS)-guided biopsy [7,8]. Imaging modalities play a critical role in both the detection of non-invasive tumors and in guiding histopathological assessment, particularly in the evaluation of metastatic spread prior to treatment or in cases of biochemical recurrence (BCR) [2]. Conventional anatomical imaging techniques, including magnetic resonance imaging (MRI) and computed tomography (CT), offer detailed structural evidence [2,9]. However, their sensitivity in detecting tumors and metastases with a size of <0.5 cm is limited, and they cannot reliably differentiate benign from malignant lesions, because they lack functional cellular information [10,11].

Molecular imaging complements anatomical imaging by providing insights into cellular and molecular biology. Functional imaging, which visualizes biological processes based on molecular biomarkers, has emerged as a valuable tool in early disease detection and treatment planning [9,10]. This concept is integral to radiotheranostics, which combines molecular imaging with targeted therapy, allowing for patient selection based on tracer biodistribution and treatment response prediction [12]. Therefore, understanding the molecular pathophysiology of radiotracers is essential for the accurate interpretation of functional imaging and its application in targeted therapy.

The primary molecular imaging modalities currently in clinical use include positron emission tomography (PET) and single-photon emission computed tomography (SPECT), which are often integrated with anatomical imaging [9,10]. These techniques provide functional insights based on molecular targeting, enhancing diagnostic precision. Although fluoro-2-deoxyglucose (FDG) PET/CT has become a cornerstone in oncologic imaging, it has limited utility in PCa because of the typically low glucose metabolism of well-differentiated tumors. FDG PET/CT is primarily reserved for poorly differentiated PCa, recurrence assessment, and prognostication. In response to these limitations, novel radiotracers have been developed for improved PCa detection, localization, staging, and treatment planning. These novel radiotracers include choline-based tracers, amino acid analogs, and prostate-specific membrane antigen (PSMA)-targeting agents, which have revolutionized molecular imaging and targeted radionuclide therapy [13,14,15].

This review aims to explore the molecular mechanisms underlying PCa and discuss the current nuclear medicine radioligands employed in both diagnostic and therapeutic settings. Additionally, we highlight their clinical applications and future perspectives, focusing on emerging radiotracers with the potential to further advance PCa management.

## 2. Nuclear Medicine Approaches for the Diagnosis and Staging of PCa

The initial diagnostic evaluation of PCa is guided by a combination of PSA testing, DRE, and imaging modalities, including ultrasonography, CT, and MR [2,16]. TRUS serves as the first-line imaging tool, offering real-time visualization and facilitating targeted biopsies [2,4]. MRI is considered the most accurate and non-invasive imaging modality for PCa diagnosis, with a crucial role in assessing the extracapsular extension of the primary tumor and pelvic lymph node involvement. The Prostate Imaging Reporting and Data System (PI-RADS) scoring system provides a standardized approach to the interpretation of MRI data, which correlates well with the Gleason score and offers prognostic significance [2,4,17]. Abdominopelvic CT is commonly employed for staging, allowing for the evaluation of extracapsular tumor extension, nodal metastasis, and visceral involvement [2,5]. Along with anatomical imaging, functional imaging techniques are integral to comprehensive staging and therapeutic planning. Pre-treatment imaging may include bone scintigraphy, PET/CT, PET/MRI, and advanced MRI techniques, such as magnetic resonance spectroscopy [4,5]. Hybrid imaging modalities, including SPECT/CT and PET/CT, combine anatomical and functional information, providing enhanced diagnostic accuracy. In particular, PET/CT offers a whole-body, non-invasive evaluation of primary tumors, regional lymph nodes, and distant metastases in a single imaging session [18]. Over the years, various radionuclides have been developed for clinical application, with the introduction of innovative, tumor-specific tracers that offer both diagnostic and therapeutic potentials. Figure 1 depicts the current theranostic methodologies in the treatment and management of patients with PCa.

## 3. Function of Radiotracers in PCa Imaging

This section provides a comprehensive review of the radiotracers used in PCa imaging, focusing on their clinical indications, mechanisms of action, and comparative efficacy across different disease stages. The selection of an appropriate imaging modality is critical for enhancing diagnostic accuracy, guiding treatment strategies, and monitoring disease progression or recurrence.

Nuclear medicine techniques, including PET/CT and SPECT, are essential for detecting localized and metastatic PCa, particularly in situations where conventional imaging methods are limited. PSMA-targeted radiotracers have demonstrated superior sensitivity, especially in patients with BCR and low PSA levels. In addition, bone-specific and metabolic radiotracers remain valuable for evaluating skeletal metastases and assessing the treatment response. The advantages and limitations of each radiotracer, along with their sensitivity, specificity, and clinical applications, are summarized in Table 1.

### 3.1. Radiopharmaceuticals for Bone Imaging

The National Comprehensive Cancer Network (NCCN) guidelines recommend bone imaging for high-risk PCa patients during the initial evaluation, in cases of elevated PSA levels following radical prostatectomy or radiation therapy, and for monitoring metastatic disease progression [5,19]. Technetium-labeled bone scintigraphy agents, including methylene diphosphonate and hydroxymethylene diphosphonate, are widely utilized in clinical practice because of their high sensitivity [4,5]. In cases of equivocal findings on bone scintigraphy, advanced imaging modalities, including F-18 sodium fluoride (NaF) PET, C-11 choline PET, and PSMA PET/CT or PET/MR, may be considered for improving diagnostic accuracy [2,5].

#### 3.1.1. Technetium-99m (Tc-99m) Bone Scintigraphy and SPECT/CT

Tc-99m bone scintigraphy is a commonly used imaging modality for detecting skeletal metastases in clinical practice. Although planar imaging is standard, SPECT/CT, a 3D hybrid technique, offers enhanced diagnostic accuracy by providing higher resolution and more accurate anatomical localization of the regions of interest [4]. Diphosphonates used in bone scintigraphy bind to hydroxyapatite in the bone through chemisorption, with uptake increasing in areas with an elevated bone turnover, such as those affected by bone metastases [2,13].

Although Tc-99m bone scintigraphy has a high negative predictive value, its positive predictive value is lower due to the tracer’s non-tumor-specific nature [13,20]. Additionally, reliable detection of metastatic bone lesions often requires PSA levels of >20 ng/mL [13]. Given these limitations, PSMA ligands are emerging as a potential substitute for bone scintigraphy, offering potentially higher sensitivity and specificity for detecting bone metastases in patients with PCa.

#### 3.1.2. PET/CT for Skeletal Images

##### F-18 NaF PET/CT

F-18 NaF PET/CT offers superior sensitivity and specificity as compared with conventional Tc-99m bone scintigraphy due to its higher resolution and the advantages of 3D whole-body hybrid imaging with CT. F-18 NaF PET/CT has been reported to be comparable to F-18 PSMA piflufolastat (DCFPyL) in detecting bone metastases [13,21]. One of its key advantages is a shorter uptake time, typically approximately 1 h as compared to conventional bone scintigraphy with an uptake time of 4 h. The pharmacokinetics of F-18 NaF involve chemisorption, where the fluoride ions bind to the surface of hydroxyapatite, similar to diphosphonate scintigraphy [13]. Despite its advantages, F-18 NaF PET/CT has not widely replaced conventional bone scintigraphy due to cost-effectiveness concerns [4,5].

##### C-11 Choline PET/CT

Biochemical analyses have demonstrated increased choline kinase activity in tumor cells, including PCa. Elevated kinase expression leads to enhanced choline uptake, which is used to synthesize phosphatidylcholine and other choline-derived membrane components [2,22,23]. High concentrations of phosphatidylcholine can be detected using radiolabeled choline, which has shown good specificity for assessing bone metastasis [2]. The NCCN guidelines recommend considering C-11 choline PET scans for evaluating skeletal metastases when initial bone scintigraphy results are equivocal [5]. However, the clinical use of C-11 choline PET is limited due to its short half-life (20 min), requiring an on-site cyclotron, and its lower sensitivity as compared with PSMA-binding radioligand imaging [2]. As a result, C-11 choline PET has not been widely adopted in clinical practice.

### 3.2. Radiopharmaceuticals for Soft Tissue Imaging

#### 3.2.1. F-18 FDG PET/CT

FDG, a glucose analog, accumulates in tumor cells due to enhanced glycolysis, a hallmark of malignant cells [24,25]. Although FDG PET/CT plays a key role in the imaging of various neoplasms, it is not commonly used for PCa because of its relatively low metabolic activity and the high urinary activity from renal excretion, which can interfere with the evaluation of prostatic lesions [24,26]. The proximity of the bladder to the prostate also increases the risk of false positives, particularly with benign conditions such as benign prostatic hyperplasia, prostatitis, and cystic malformations [27,28]. As a result, the NCCN guidelines do not recommend the routine use of FDG PET/CT for the initial staging of PCa. However, it may be considered in select cases, particularly those with a Gleason score > 7, high-grade hormone-resistant disease, poorly differentiated lesions, or for prognostic evaluation [4,5]. Additionally, tumor differentiation is inversely correlated with FDG uptake—higher FDG avidity is generally associated with a poorer prognosis compared with lower FDG uptake [29,30,31].

#### 3.2.2. F-18 Fluorocholine (FCH) PET/CT

F-18 FCH PET/CT offers a significant advantage over C-11 choline PET/CT due to its longer half-life, making it more widely available for clinical use (Figure 2). Unlike C-11 choline, which is primarily used for bone metastasis detection, F-18 FCH has been approved for evaluating both soft tissues in cases of BCR or disease progression and for the early detection of bone metastasis, particularly when serum PSA levels exceed 1.0 ng/mL [5,32,33]. The increased choline metabolism observed in tumor cells is linked to enhanced phospholipid membrane synthesis and cell proliferation, which are indicative of biological aggressiveness in PCa [34,35]. F-18 FCH has demonstrated superior avidity compared with FDG in both androgen-dependent and androgen-independent PCa [36,37]. Despite urinary excretion, F-18 FCH PET/CT shows promise as a highly specific modality for primary tumor evaluation [38]. However, its clinical utility in PCa is limited by its low sensitivity, variable detection performance, and lower sensitivity as compared with that of PSMA-binding radioligand imaging [39,40,41,42]. The advent of F-18 PSMA-1007, which is excreted via the hepatobiliary system, has further reduced the role of F-18 FCH in clinical practice [5,43].

#### 3.2.3. Fluciclovine (FACBC) PET/CT

F-18 FACBC is a radiolabeled amino acid leucine analog utilized in PCa imaging due to the increased amino acid transport and metabolism observed in malignant cells [2,13]. Initially, it was anticipated that its hepatic excretion would enhance lesion visualization near the bladder. However, despite offering superior detection compared with CT alone, F-18 FACBC has demonstrated limited specificity in primary PCa, nodal diseases, and recurrent lesions. Consequently, its clinical utility has been primarily recommended for cases of BCR, disease progression, and equivocal findings in bone scintigraphy, rather than for initial staging [13]. Due to its lower sensitivity compared with that of PSMA-targeted tracers, F-18 FACBC has since been removed from the NCCN guidelines [5,13].

#### 3.2.4. PSMA

PSMA is a transmembrane glycoprotein that is overexpressed in PCa cells, with the expression levels being correlated with tumor aggressiveness, including castration-resistant and metastatic disease [13,44,45]. Given its minimal expression in normal prostate and non-prostatic tissues, PSMA has emerged as a highly appealing target for radiopharmaceuticals in diagnostic, prognostic, and therapeutic settings [13,44]. Notably, approximately 5% of PCa cases are PSMA-negative, which is often associated with poor prognostic outcomes [13,46,47,48]. The most recent NCCN guidelines have recommended PSMA-PET as a superior front-line imaging modality due to its high sensitivity and specificity in detecting micrometastases, outperforming conventional imaging techniques, including CT, MRI, and bone scintigraphy, in both initial staging and disease recurrence/progression [5]. PSMA-targeted radioligands can be broadly categorized as follows on the basis of their mechanism of action: monoclonal antibodies targeting the extracellular domain and small-molecule ligands binding to the intracellular domain, facilitating internalization [44]. Table 2 shows a comprehensive summary of the clinically relevant PSMA radioligands.

#### 3.2.5. Monoclonal Antibodies

The first PSMA-targeted radioligand introduced for clinical use was 7E11-C35, a monoclonal antibody radiolabeled with In-111 (In-111 capromab pendetide; Prostascint). However, its clinical utility was limited due to its relatively poor sensitivity at low PSA levels and lower resolution in SPECT imaging [42,45]. To improve specificity, the humanized monoclonal IgG1 antibody, huJ591, labeled with the positron emitter Zr-89 was developed. Despite its enhanced specificity, huJ591 demonstrated a slow plasma clearance and delayed tumor uptake [42,45]. To address these challenges, Zr-89 Df-IAB2M was designed to achieve faster blood clearance. However, its application to the clinical field is limited due to the prolonged interval between injection and imaging, which is approximately 48 h [42,45,49].

#### 3.2.6. Small-Molecule Ligands

Ga-68 PSMA-HBED-CC (Ga-68 PSMA-11) is the most extensively utilized PET radiotracer for PSMA-targeted imaging [42]. The Society of Nuclear Medicine and Molecular Imaging and the European Association of Nuclear Medicine have updated the clinical indications for PSMA-ligand PET/CT, which now include the initial staging of intermediate-to high-risk PCa, localization of metastatic disease in BCR or persistent PCa cases, and response assessment before and after RLT [50,51].

F-18-labeled radioligands have been investigated as alternatives to Ga-68-based agents, offering advantages, including greater availability and improved image resolution. F-18 DCFPyL has demonstrated a strong diagnostic performance in both the primary staging and restaging of PCa [52,53]. F-18 florastamin (F-18 FC303) is an emerging PSMA-targeted diagnostic radioligand that exhibits a higher tumor-to-background contrast as compared to F-18 DCFPyL, facilitating superior detection of osseous metastases, small metastatic lymph nodes, and primary tumors on delayed imaging [54]. Additionally, F-18 PSMA-1007 has been developed to minimize urinary excretion, enhancing the detection rates at lower PSA levels in both BCR and initial staging [42,55,56] (Figure 2). More recently, F-18 F-rhPSMA-7.3, which has received FDA approval, has been extensively evaluated and has demonstrated excellent diagnostic accuracy for nodal staging in intermediate- to high-risk PCa patients (Figure 3) [57].

While PSMA PET/CT has emerged as a valuable imaging modality for PCa, particularly in initial staging, detection of biochemical relapse, and guidance of targeted therapy, its widespread clinical adoption remains limited. Key barriers include restricted access to PET equipment, requirements for radiation safety infrastructure, and high operational costs—especially in remote or resource-limited settings [58,59,60]. In contrast, SPECT is more widely available globally. Tc-99m-labeled radiotracers, commonly used in SPECT imaging, are cost-effective, readily accessible, simple to produce, and have a long half-life, making them practical for routine nuclear medicine use [58,61,62,63,64]. Technological advancements in SPECT have also narrowed the gap in image resolution between SPECT and PET, further enhancing its clinical utility [60,65,66].

A recent comparison study highlighted the potential of Tc-99m PSMA as a cost-effective substitute for Ga-68 PSMA PET imaging in the diagnostic evaluation of advanced-stage PCa. This is particularly relevant in the majority of healthcare settings, where access to PSMA PET remains limited, with only a few specialized centers equipped for such imaging. In this scenario, patients with extensive metastatic disease who are candidates for PSMA therapy at tertiary care facilities can undergo baseline and follow-up imaging with PSMA at peripheral centers that lack PET infrastructure but are equipped with conventional nuclear medicine facilities [67]. Supporting this approach, Sergieva et al. reported encouraging results using Tc-99m PSMA-T4 in the detection of PCa recurrence [68]. They emphasized the utility of this SPECT tracer in identifying PSMA expression prior to initiating RLT, as well as in monitoring the response to treatment.

Currently, several Tc-99m-labeled PSMA derivatives have been developed and evaluated for SPECT imaging, including Tc-99m MIP-1404, Tc-99m HYNIC-Glu-Urea-A, Tc-99m mas3-ynal-k(Sub-KuE), Tc-99m PSMA-T4, Tc-99m PSMA-11, Tc-99m HYNIC-PSMA, and Tc-99m PSMA-I&S. These radiotracers have demonstrated excellent diagnostic performance, with high sensitivity and specificity for detecting both primary tumors and recurrent disease in PCa patients [58,60]. Although none of these agents have yet received FDA approval, their promising clinical results position Tc-99m-labeled PSMA compounds as strong candidates for broader adoption in standard PCa imaging protocols [60,69].

While the detection of additional metastatic lesions through advanced imaging techniques may enhance diagnostic confidence, it does not necessarily translate into significant clinical benefit. In routine practice, the therapeutic strategy for PCa patients with metastatic disease typically remains unchanged despite the identification of more lesions—unless one modality fails to detect any metastases at all. In such cases, a negative result may influence decisions regarding eligibility for systemic therapies or RLT. Therefore, the clinical utility of a given imaging modality should not be judged solely on its ability to identify more metastatic sites, but rather on whether it provides actionable information that alters patient management [67,70].

## 4. The Role of Target Therapy in Nuclear Medicine in PCa Management

BCR in PCa is defined as an increasing serum PSA level following a definitive treatment. It occurs in approximately 35% of patients within 10 years after radical prostatectomy [71,72]. Recurrent or progressive cases are typically managed with androgen deprivation therapy (ADT); however, within 2 years, many patients develop CRPC, a stage in which the disease no longer responds to hormonal therapy [73,74,75]. At this point, PCa often metastasizes rapidly, leading to symptomatic disease progression. Although chemotherapy remains the standard treatment for CRPC, the overall survival (OS) benefits are limited [71,75].

RLT has emerged as a promising alternative in this setting, offering both therapeutic and imaging capabilities through theranostic radiopharmaceuticals. By combining diagnostic imaging with treatment, these agents enable precise targeting, allowing for patient selection, treatment monitoring, and dose optimization while minimizing adverse effects. The theranostic approach provides several advantages, including real-time visualization of drug biodistribution, improved patient stratification to reduce the risk of non-responders, and enhanced treatment response assessment [12].

The following section will explore the targeted radiopharmaceuticals in nuclear medicine, emphasizing their dual role in imaging and therapy, with a focus on precision-targeted approaches. Table 3 shows the current radiopharmaceuticals used in treatment settings.

### 4.1. Targeted RLT in PCa

Targeted therapy aims to selectively destroy the tumor cells while sparing the normal tissues [42]. Given PCa’s radiosensitivity, RLT has emerged as a promising treatment option. Historically, β-emitting radiopharmaceuticals, including Re-186, Sr-89, and Sm-153, have been used for skeletal metastases. However, Ra-223, an α-emitting radiopharmaceutical, has become the first-line treatment for symptomatic bone metastases or bone-predominant cases following chemotherapy in the absence of visceral metastases [76].

Ra-223 is the first targeted α therapy approved for metastatic castration-resistant prostate cancer (mCRPC). It has a half-life of 11.4 days and binds to the bone mineral hydroxyapatite in areas of high bone turnover, delivering high linear energy transfer (LET) radiation directly to tumor cells, leading to double-stranded DNA breaks [77,78,79]. Given its short tissue penetration range (<100 µm), Ra-223 maximizes tumor cell destruction while minimizing toxicity to the bone marrow and surrounding healthy tissues, making it superior to β-emitters, including Sm-153 or Sr-89 [77,80]. Clinical trials have demonstrated that Ra-223 considerably improves the OS and maintains the quality of life in patients with bone-predominant mCRPC [77,81,82,83,84].

Despite its efficacy, Ra-223 has several clinical limitations. Its use is restricted to symptomatic bone-predominant disease post-chemotherapy, and it cannot be used in patients with visceral metastases. Furthermore, its administration requires specialized radiation facilities, strict dose-limited control areas, and radiation safety protocols for healthcare providers [2,79]. These constraints have driven the exploration of alternative radiopharmaceuticals to expand the treatment options.

Ac-225 PSMA-targeted therapy is a promising α-emitting radioligand targeting PSMA, which is overexpressed in mCRPC. With higher LET as compared to β-particles, Ac-225 is more effective at inducing double-stranded DNA breaks, leading to potent tumor cell killing [85]. In Sathekge et al.’s study, 73% of their patients exhibited a decline in the PSA level following at least one cycle of Ac-225 PSMA RLT [85]. Parida et al. also reported that 60% of their patients experienced a PSA level reduction of >50% [86]. Additionally, Ac-225 PSMA RLT has demonstrated prolonged tumor control and improved OS, even in patients with progressive disease, following Lu-177 PSMA therapy [85]. These findings suggest that Ac-225 PSMA RLT could serve as a viable last-line therapeutic option for refractory mCRPC [85]. However, its clinical implementation remains challenging due to limited production capacity, the presence of short-lived α- and β-emitting decay products, and the lack of established imaging methods for treatment monitoring [87].

As research continues, optimizing the targeted radiopharmaceuticals, refining patient selection, and improving accessibility will be critical for integrating RLT into broader clinical practice.

### 4.2. Lu-177 PSMA-Targeted Therapy in PCa

Lu-177 is the cornerstone of theranostics in PCa treatment, offering both therapeutic and diagnostic capabilities [2,88]. As a high-energy β-emitting radionuclide with a short tissue penetration range (maximum of 2 mm), it minimizes collateral damage to the surrounding healthy tissues [13,89]. Additionally, its low-energy gamma emission allows imaging with gamma cameras, facilitating personalized dosimetry and treatment planning [13]. Lu-177 is commonly used in combination with PSMA-binding ligands, specifically targeting PSMA-overexpressing tumors. Lu-177 PSMA-617, the first-generation systemic RLT, has been approved by the FDA for patients with progressive mCRPC who are intolerant to ADT and taxane-based chemotherapy [90]. The eligibility for treatment requires adequate PSMA expression on a baseline PSMA PET/CT scan [13].

Clinical trials have demonstrated that Lu-177 PSMA-617 considerably improves patient survival, enhances the quality of life by delaying skeletal events, provides superior pain control, and is better tolerated than chemotherapy [71,91]. The primary principle of RLT is to deliver the maximum dose to the target tissues while minimizing exposure to healthy organs. The main organs affected by Lu-177 PSMA-617 include the parotid glands, kidneys, and bone marrow [90,92]. Although the toxicities are generally mild, xerostomia due to salivary gland involvement occurs in up to 30% of patients and can negatively impact their quality of life [92]. To address these limitations, second-generation PSMA-based radiopharmaceuticals, including Lu-177 Ludotadipep, have been developed. Based on the glutamate–urea–lysine (GUL) structure, Lu-177 Ludotadipep is designed to extend the circulation time by binding to albumin, reducing non-specific organ uptake, increasing total tumor absorption, and lowering the side effects with smaller doses (Figure 4) [71,93,94].

A clinical trial (NCT05458544) is currently evaluating Lu-177 Ludotadipep in two phases. Phase 1 focuses on safety, tolerability, and determining the maximum tolerated dose (MTD) and recommended Phase 2 dose through a single administration of 3.7 GBq (or 2.775 GBq if necessary). This phase includes patient screening, imaging, and protective measures for the salivary glands and kidneys, followed by an 8-week monitoring period for adverse events and dose-limiting toxicities. Phase 2a assesses the safety and efficacy of repeated doses every 8 weeks (4–6 cycles), with continuous monitoring through imaging, laboratory tests, and safety assessments. A long-term follow-up extending up to 10 years will evaluate the potential late-onset complications, including myelosuppression and secondary malignancies. These advancements in PSMA-targeted RLT highlight the evolving landscape of personalized treatment in mCRPC, optimizing therapeutic benefits while minimizing side effects.

Multiple clinical trials are ongoing to further evaluate the efficacy and optimization of Lu-177 PSMA-based treatments. The following section summarizes the key clinical trials investigating the role of Lu-177 PSMA therapy in advanced PCa.

## 5. Current Clinical Trials and Future Perspectives

Defining the role of RLT in the clinical setting of PCa is a continuing and ongoing process. Lu-177 PSMA-617 has notable efficacy in the treatment of mCRPC. Evidence of its effectiveness includes reductions in the PSA levels and tumor size, along with considerable pain relief and a tolerable side-effect profile [95]. While excluding treatments, such as chemotherapy, immunotherapy, Ra-223, and investigational drugs, clinical trials have shown that incorporating Lu-177 PSMA-617 into protocol-permitted standard care prolongs imaging-based progression-free survival (PFS) and OS when compared with standard care alone in patients with advanced PSMA-positive mCRPC [96]. RLT is evolving into a promising treatment option for mCPRC patients. Recently, PSMA has been recognized as a potential theranostic target for the treatment of mCRPC.

### Current and Ongoing Clinical Trials

The TheraP clinical study is a multicenter randomized phase II trial that evaluated a group of 200 men with CRPC who were eligible for cabazitaxel as their next standard treatment [91]. The participants were allowed to undergo prior treatment with androgen receptor-targeted therapies. As part of the study, all patients underwent imaging with PSMA PET/CT and FDG PET/CT. This study included only FDG-positive but PSMA-negative lesions.

The patients were randomly assigned to one of the following two treatment arms: the first group receiving Lu-177 PSMA-617 (6.0–8.5 GBq administered intravenously every 6 weeks for up to six cycles), and the second group receiving cabazitaxel (20 mg/m^2^ intravenously every 3 weeks for up to ten cycles). The trial’s primary endpoint was a reduction in PSA levels by at least 50% from baseline. This endpoint was achieved significantly more often in the Lu-177 PSMA-617 group than in the cabazitaxel group. PSA responses were observed in 66% of patients receiving Lu-177 PSMA-617 and in 37% of those receiving cabazitaxel according to the intention-to-treat analysis (*p* < 0.0001). Similarly, in the treatment-received analysis, the response rates were 66% and 44% for the Lu-177 PSMA-617 and cabazitaxel groups, respectively (*p* = 0.0016).

In addition to demonstrating superior efficacy, the Lu-177 PSMA-617 group reported fewer severe adverse events (grade 3 or 4), highlighting its more favorable safety profile. Despite these differences, the OS was similar between the two groups. Over a 36-month follow-up period, the restricted mean survival times were 19.1 and 19.6 months for the Lu-177 PSMA-617 and cabazitaxel groups, respectively, with a mean difference of −0.5 months (95% confidence interval [CI]: −3.7 to 2.7). These results underscore the potential of Lu-177 PSMA-617 as an effective and well-tolerated treatment option for advanced CRPC [97].

The VISION trial has evaluated the efficacy of Lu-177 PSMA-617, an innovative RLT, in mCRPC patients. This study enrolled 831 participants who had previously received at least one androgen receptor pathway inhibitor and one or two taxane-based chemotherapy regimens. The eligibility required PSMA-positive disease, confirmed through gallium-68 (Ga-68)-labeled PSMA-11 PET/CT scans. The patients were excluded if their scans showed PSMA PET-negative visceral or lytic bone metastases measuring ≥ 1 cm, or lymph nodes ≥ 2.5 cm, as these could indicate resistance to the therapy. The participants were randomized in a 2:1 ratio to receive either Lu-177 PSMA-617 (7.4 GBq every 6 weeks for four to six cycles) with protocol-permitted standard care or standard care alone. Importantly, the protocol-permitted standard care excluded treatments such as cytotoxic chemotherapy, systemic radioisotopes, immunotherapy, or investigational drugs, such as olaparib, because of limited safety data on their combination with Lu-177 PSMA-617. Although these exclusions ensured patient safety, they also differed from the standard-of-care treatments, which include therapies such as chemotherapy and Ra-223 that are known to extend survival in certain mCRPC patients.

The trial produced considerable results, achieving its primary endpoints of extended imaging-based PFS and OS. The Lu-177 PSMA-617 group had a median PFS of 8.7 months, whereas that of the control group was 3.4 months (hazard ratio [HR] for progression or death: 0.40; 99.2% CI: 0.29–0.57; *p* < 0.001). The median OS was also markedly improved at 15.3 and 11.3 months in the Lu-177 PSMA-617 and standard-care-alone groups, respectively (HR for death: 0.62; 95% CI: 0.52–0.74; *p* < 0.001). These findings reinforced the potential of Lu-177 PSMA-617 as a life-prolonging therapy in advanced PCa.

Although effective, Lu-177 PSMA-617 was associated with a higher rate of grade ≥ 3 adverse events as compared with standard care alone. Nonetheless, it demonstrated additional clinical benefits, including a prolonged time to symptomatic skeletal events [11.5 months vs. 6.8 months in the control group; HR: 0.50; 95% CI: 0.40–0.62; *p* < 0.001)]. Moreover, the therapy delayed the worsening of health-related quality of life (HRQoL) and pain, as measured by established tools such as the Functional Assessment of Cancer Therapy-Prostate, Brief Pain Inventory, and Short Form questionnaires [98].

In conclusion, the VISION trial highlighted the promise of Lu-177 PSMA-617 in extending survival and improving the quality of life of mCRPC patients. Despite its higher incidence of adverse events, its clinical benefits make it a valuable option in the treatment landscape of advanced PCa.

The ALpharadin in SYMptomatic Prostate Cancer Patients trial enrolled 921 individuals diagnosed with symptomatic CRPC. These patients had at least two symptomatic bone metastases, no known visceral metastases, and were receiving the best standard of care available. The participants were randomized into the following two groups: a group receiving six injections of Ra-223 dichloride—a novel, targeted α-emitting radiopharmaceutical designed to selectively bind to areas of heightened metabolic activity in bone metastases—administered intravenously at a dose of 50 kBq/kg every 4 weeks, and another group receiving a matching placebo [99].

The key results from the trial demonstrated several considerable benefits associated with Ra-223. The notable findings included pain relief, a prolongation of median OS (regardless of prior treatment with docetaxel), and a reduction in the risk of symptomatic skeletal events among patients who had not previously received docetaxel. Specifically, for patients with prior docetaxel exposure, the HR for OS was 0.70 (95% CI, 0.56–0.88; *p* = 0.002), whereas for those without prior docetaxel treatment, the HR was 0.69 (95% CI, 0.52–0.92; *p* = 0.01). These results highlighted Ra-223 as a promising therapeutic option for managing bone metastases in this patient population [83].

However, when Ra-223 was later evaluated in combination with abiraterone acetate in the ERA 223 trial, the outcomes were less favorable. The combination therapy did not demonstrate an improvement in OS as compared with abiraterone alone. Moreover, it was associated with a higher incidence of non-cancer-related skeletal events. The underlying cause of this adverse outcome remains unclear, but it suggests the potential interaction between abiraterone and Ra-223, possibly interfering with the radiopharmaceutical’s ability to bind to areas of increased metabolic activity in bone metastases. This unexpected result underscores the complexity of combining radioligands with other therapeutic agents and serves as a cautionary example when designing combination therapy regimens involving radiopharmaceuticals [100]. The clinical trials are summarized in Table 4.

## 6. Future Directions of Nuclear Medicine in the Clinical Setting of PCa

Advances and the development of novel radiotheranostic procedures are actively ongoing. Progress can be observed in various fields and venues, including combination regimens, radionuclides, different platforms, and alternative approaches for PCa patients in the clinical setting of nuclear medicine.

### 6.1. Combination Therapies in Radiotheranostics: Expanding the Therapeutic Horizons for Clinicians

The integration of combination therapies with radiotheranostics represents an important advancement in oncology, offering clinicians new tools to enhance therapeutic efficacy and improve clinical outcomes. By leveraging the synergistic effects of RLTs with other treatment modalities, these combinations have the potential to overcome resistance mechanisms, broaden treatment indications, and reduce adverse effects through optimized dosing strategies. The integration of combination therapies in PSMA-targeted RLT is gaining momentum as a strategy to enhance treatment efficacy and overcome disease progression. Given that Lu-177 PSMA therapy alone can extend survival in mCRPC patients but does not consistently prevent disease progression—particularly to the bone and liver—multimodal treatment strategies have been increasingly evaluated [101].

For mCRPC, combining RLT with androgen receptor signaling inhibitors is an innovative strategy, given that these agents may upregulate PSMA expression, thereby enhancing the effectiveness of PSMA-targeted RLT. The Enza-P trial is currently evaluating the combination of enzalutamide and Lu-177 PSMA-617 in mCRPC patients who have progressed on docetaxel but are naive to treatment with androgen receptor inhibitors [102]. Clinicians should note that the primary endpoint is PSA PFS, with secondary endpoints including radiological PFS, OS, pain response, adverse events, and cost-effectiveness—providing a comprehensive assessment of the treatment’s clinical benefit.

The combination of RLT with DNA repair inhibitors, which could enhance the cytotoxic effect by exploiting the defects in the DNA damage response pathways, is another area of active investigation. The LuPARP trial is a phase I, open-label, multicenter study evaluating the combination of Lu-177 PSMA-617 and olaparib, a PARP inhibitor, in mCRPC patients who have progressed on androgen receptor-targeted therapies. This two-phase study included dose escalation followed by an expansion phase to establish the recommended phase II dose. Preliminary safety data from this trial will inform the design of subsequent phase II/III studies and may offer clinicians a new therapeutic option for heavily pretreated patients [103].

The LuPIN trial explored the combination of Lu-177 PSMA with NOX66, a flavonoid-derived agent designed to enhance apoptotic signaling via mitochondrial caspase activation. This approach was tested in end-stage mCRPC patients to increase tumor radiosensitivity and improve treatment outcomes. The study demonstrated a PSA response rate of >50% in 63% of patients with mCRPC who had previously been treated with taxane chemotherapy and androgen receptor inhibitors. Clinicians should be aware of potential side effects, including anal inflammation related to idronoxil’s delivery method, although other adverse events were consistent with those expected for Lu-177 PSMA-617 [104].

The combination of RLT with immune checkpoint inhibitors is an area of growing interest, particularly in treatment-resistant PCa. The PRINCE trial is a phase Ib/II study evaluating the combination of Lu-177 PSMA-617 and pembrolizumab in mCRPC patients. The initial results are promising, with a PSA50 response rate of 76%, median PFS of 11.2 months, and OS of 17.8 months. However, the absence of a comparator arm with Lu-177 PSMA-617 monotherapy limits the study’s ability to determine the precise contribution of pembrolizumab to these outcomes [105].

Combination therapies involving radioligand agents and cytotoxic chemotherapy are also demonstrating considerable clinical promise. A phase I study combining Lu-177 J591 with docetaxel in mCRPC patients reported a PSA decline of >50% in 73.3% of patients and a reduction in circulating tumor cell counts in 85.7% [106]. The UpFrontPSMA trial has evaluated a similar combination treatment, comprising Lu-177 PSMA-617 and docetaxel [107].

Bone metastases remain a major site of disease progression in mCRPC, often leading to skeletal-related complications and limiting OS. Recognizing this challenge, the AlphaBet trial [108] investigated the combination of Ra-223 and Lu-177 PSMA to target both bone and soft-tissue metastases. Ra-223, an α-emitting radiopharmaceutical, selectively deposits radiation in areas of active bone turnover, making it an attractive candidate for treating micrometastatic bone disease. By combining Ra-223 with Lu-177 PSMA, bone progression is delayed or prevented while maintaining low marrow irradiation, thereby minimizing hematologic toxicity. Additionally, the DORA trial has investigated the combination of Ra-223 and docetaxel in mCRPC, with the aim to improve survival outcomes and provide an alternative approach for patients who developed resistance to single-agent therapies [109].

Although combination therapies are often considered salvage options when monotherapy fails, their potential extends beyond the salvage treatment alone. A proactive multimodal approach—integrating RLT with systemic agents, including androgen receptor pathway inhibitors, DNA damage response inhibitors, or immune checkpoint blockade—could optimize disease control, delay resistance mechanisms, and potentially improve survival. Therefore, future research should focus on refining the patient selection criteria, optimizing treatment sequencing, and assessing long-term toxicity profiles to maximize the clinical benefit of these combination strategies. The clinical trials of combination therapy with RLT are summarized in Table 5.

### 6.2. Advancements in PSMA-Targeted RLT: Emerging Radionuclides and Chelator Technologies

The evolving landscape of PSMA-targeted RLT is marked not only by the development of new PSMA ligands but also by the introduction of innovative radionuclides and advanced chelator technologies. These advancements aim to optimize imaging and therapeutic efficacies while enhancing the stability and safety profile of PSMA-based theranostics. Among these emerging approaches, copper and scandium radioisotopes and novel chelators hold particular promise for both diagnostic and therapeutic applications in PCa and other malignancies.

Copper radioisotopes, including Copper-64 (Cu-64) and Copper-67 (Cu-67), are re-emerging as versatile agents for theranostic applications. Cu-64 is inherently theranostics, allowing for both PET imaging and targeted radiotherapy, while also pairing uniquely with Cu-67 for a broader therapeutic window. However, preliminary clinical evaluations of Cu-64 PSMA-617 in PCa patients have revealed suboptimal complex stability, with notable radiotracer accumulation in the liver, potentially limiting its clinical utility [110]. This hepatic uptake underscores the need for improved chelation strategies to enhance in vivo stability and reduce off-target toxicity.

In response, research has focused on the development of PSMA constructs incorporating albumin-binding groups, which not only enhance the circulation time but also improve the tumor uptake. One such construct, Cu-64 PSMA-ALB-89, utilizes a NODAGA-containing analog of PSMA-ALB-56 and has demonstrated promising accumulation in PC3-PIP xenograft models, albeit with considerable renal uptake [111]. Contrarily, Cu-64 RPS-085, a derivative of RPS-063 featuring a sarcophagine chelator, exhibits more favorable pharmacokinetics with faster renal clearance, highlighting the benefits of the sarcophagine moiety in achieving high chelation stability [112]. Compared to other copper-labeled radioligands, particularly Cu-64 PSMA-617, constructs such as Cu-64 RPS-085 and the bivalent sarcophagine inhibitor Cu-64 SAR-bisPSMA demonstrate minimal hepatic activity, which is indicative of the enhanced stability provided by sarcophagine chelation [113].

The therapeutic potential of sarcophagine chelators is further evidenced by ongoing clinical trials evaluating sarcophagine-conjugated octreotate molecules for imaging and targeted RLT in neuroendocrine cancers [114]. Additionally, preclinical studies have shown that a single 30-MBq dose of Cu-67 SAR-bisPSMA can achieve tumor control for up to 6 weeks [114], suggesting robust anti-tumor activity and highlighting sarcophagine chelators as a promising tool in PSMA-based theranostics.

The theranostic triad of scandium isotopes, namely, Sc-43, Sc-44, and Sc-47, also holds marked potential for the clinical management of PCa. These isotopes are particularly well-suited for radioligands with extended circulation times, such as albumin-binding PSMA ligands, and may offer advantages over traditional agents, such as Ga-68. Among the two positron-emitting isotopes, Sc-43 is preferred for clinical use because of the higher radiation dose associated with the high-energy gamma emission of Sc-44 [115].

Initial clinical studies on Sc-44 PSMA-617 have demonstrated PSMA-specific uptake and high complex stability, with pharmacokinetics comparable to those of Lu-177 PSMA-617 [116]. The favorable dosimetric profile observed in the first-in-human studies suggests that Sc-44 PSMA-617 could play a valuable role in pre-therapy planning and patient selection for PSMA-targeted RLT. However, the effective chelation of Sc^3+^ with DOTA requires prolonged heating at a temperature of 95 °C, presenting practical challenges for clinical application. To address this, new chelators such as picaga have been developed, facilitating the rapid and stable complexation of Sc^3+^ and Lu^3+^ under milder conditions. Preclinical studies with Lu-177 picaga-Alb53-PSMA have shown promising tumor growth inhibition for over 3 weeks, indicating that optimized chelation strategies could enhance therapeutic outcomes [117].

The expanding availability of novel radiometal β-emitters, including Sc-47 [106], Cu-67 [113,114,118], and Tb-161 [119], along with α-emitters such as Tb-149 [120], Pb-212 [121], Th-227 [122], and Ac-225, is likely to drive further innovation in PSMA-targeted RLT. The development of bifunctional chelators capable of rapid metal ion complexation with high in vivo stability is particularly critical for targeted alpha therapy, where radiometal release can lead to dose-limiting toxicities due to off-target deposition.

Although the DOTA macrocycle remains widely used, its stability with larger α-emitters, such as Ac-225, is suboptimal, as demonstrated by a decline in the complex stability with an increase in lanthanide ion size [123]. To overcome these limitations, alternative macrocyclic chelators, including macropa, crown, py-macrodipa, and mcp-D-click, have been introduced, offering enhanced stability with Ac^3+^-225 and Bi^3+^-213 [124,125,126]. Additionally, acyclic chelators, such as hydroxypyridinonate and picolinic acid derivatives, have shown potential for the stable complexation of Th^4+^-227 [127].

Overall, these innovations represent considerable advancement in PSMA theranostics, although many of the newly developed chelators remain in preclinical evaluation. The transition to clinical application will require the successful integration of these chelators into high-affinity PSMA targeting ligands, coupled with the rigorous assessment of pharmacokinetics, biodistribution, and safety profiles in human studies. As research progresses, the optimization of chelator design and radionuclide selection is anticipated to expand the therapeutic window of PSMA-targeted RLT, offering new avenues for treatment personalization and improved patient outcomes.

### 6.3. Advancements in PSMA-Targeted Radioguided Surgery for PCa Metastases

The application of PSMA-targeted radioguided surgery (RGS) represents a promising advancement in the intraoperative detection and removal of metastatic PCa lesions. This technique is particularly relevant for cases with small or atypically located metastases that are challenging to detect using conventional imaging modalities. Despite the widespread adoption of PSMA-targeted PET/CT for preoperative staging and recurrence detection, the role of PSMA-specific SPECT imaging in conjunction with intraoperative guidance remains an area of active investigation.

Since 2014, the Technical University of Munich has pioneered PSMA-RGS, initially utilizing indium-111 (In-111)-labeled PSMA ligands and later transitioning to Tc-99m-based tracers to improve its clinical applicability and imaging resolution. This innovative approach involves the intravenous administration of In-111- or Tc-99m-labeled PSMA ligands preoperatively, enabling the detection of intraoperative lesions using a gamma probe. This handheld device provides real-time acoustic and visual feedback on radioactivity levels, allowing surgeons to localize the metastatic lesions with high precision. A key advantage of this technique is the ability to confirm complete metastasis removal through an ex vivo gamma probe analysis of the excised tissue, offering immediate intraoperative validation.

Multiple studies have evaluated the clinical utility of PSMA-RGS. In Rauscher et al.’s prospective study, In-111-labeled PSMA-I&T was used in 31 patients undergoing salvage surgery for localized recurrent PCa [128]. The study reported considerable biochemical responses, with PSA declines of >50% and >90% observed in 23/30 and 16/30 patients, respectively. Additionally, 18/30 patients achieved a PSA level of <0.2 ng/mL postoperatively. Importantly, 20 out of 30 patients had not received any additional treatment at a median follow-up of 337 days (range: 81–591 days), suggesting that PSMA-RGS may contribute to prolonged treatment-free survival in select cases. In a separate retrospective study conducted by Maurer et al., Tc-99m-labeled PSMA imaging was evaluated in 31 patients with BCR following radical prostatectomy [129]. The study demonstrated a sensitivity of 83.6% (95% CI: 70.9–91.5%) and an overall accuracy of 93.0% (95% CI: 85.8–96.7%), further supporting the potential clinical value of this technique.

Despite the encouraging results, certain limitations remain. The efficacy of PSMA-RGS is inherently dependent on lesion size, with smaller metastases or micrometastases potentially evading detection due to limited tracer uptake. Moreover, although the gamma probes offer intraoperative feedback on radioactivity levels, they do not provide direct visual differentiation between tumor and normal tissue in real time. This challenge underscores the need for complementary imaging techniques or hybrid surgical approaches, including fluorescence-guided surgery, to enhance lesion visualization.

Future research should focus on optimizing radiotracer pharmacokinetics, refining gamma probe sensitivity, and integrating advanced intraoperative imaging modalities to further improve PSMA-RGS’s accuracy and clinical utility. The continued evaluation of patient selection criteria, long-term oncologic outcomes, and comparative studies against standard-of-care surgical techniques will be crucial in defining the precise role of PSMA-RGS in the management of recurrent and metastatic PCa.

### 6.4. Clinical Integration of Artificial Intelligence (AI) in PSMA Theranostics

PSMA-directed molecular imaging and drug toxicity evaluation are undergoing considerable advancements with the integration of AI and machine-learning technologies [130]. Traditionally, drug toxicity has been assessed through animal studies; however, their predictive accuracy is often limited when translated to human outcomes [131]. Previous studies have shown that toxicity predictions from animal models align with human results in only 43% of rodent studies and 63% of non-rodent studies, with <30% agreement for adverse drug reactions in the target organs [132]. This has led to a growing shift toward in vitro laboratory tests and in silicon computational models, offering a more reliable and ethical alternative. AI-based approaches, including Bayesian models, support vector machines, Random Forest, AdaBoost decision trees, Bernoulli Naive Bayes, and deep neural networks, are being extensively explored for their use in toxicity evaluation and drug–drug interaction predictions. These innovative tools could transform drug development, allowing for the earlier identification of toxic effects and improving drug safety profiles [133,134,135].

In parallel, molecular imaging is being revolutionized by machine learning and radiomics. Leung et al. demonstrated the potential of deep-learning methods in PSMA-directed molecular imaging by developing an automated approach for F-18 DCFPyL PET/CT evaluation in 207 patients [136]. Compared with conventional and semi-automated thresholding methods, deep learning provided more accurate segmentation, enabling improved therapy monitoring and long-term care planning. Machine learning and radiomics can extract multiple features from imaging data, facilitating the discovery of new biomarkers and enhancing clinical decision-making. Together, these AI-driven innovations are paving the way for more precise diagnostics, optimized RLT, and ultimately better patient outcomes.

### 6.5. Optimizing the Use of RLT Across the Treatment Lines

RLT is traditionally evaluated in patients with advanced, progressive malignancies who have undergone multiple prior treatments. A key example is Lu-177 PSMA, which is primarily used as a third-line therapy for mCRPC following treatment with an androgen receptor pathway inhibitor (ARPI) and chemotherapy with docetaxel. However, there is an increasing interest in expanding the use of these therapies into earlier treatment settings, including adjuvant and neoadjuvant applications. Although earlier-stage implementation may provide superior disease control and survival benefits, it also necessitates prolonged follow-up due to concerns regarding long-term toxicities, including secondary malignancies, renal impairment, and myelotoxicity, which are particularly relevant for patients expected to have longer post-treatment survival. Several ongoing clinical trials are investigating the efficacy and safety of RLT when introduced in earlier lines of therapy.

The PSMAddition (NCT04720157) study is a phase III trial assessing the efficacy of Lu-177 PSMA-617 in combination with standard-of-care (SOC) therapy in comparison with that of SOC alone in patients with metastatic hormone-sensitive prostate cancer (mHSPC). Eligible participants were treatment-naïve or minimally pretreated individuals with PSMA-positive disease. This study aims to determine whether the integration of Lu-177 PSMA-617 in an earlier setting can improve PFS and OS [137].

The UpFrontPSMA (NCT04343885) clinical trial focuses on patients with newly diagnosed high-volume mHSPC, as identified by PSMA PET/CT imaging. The trial is designed to evaluate whether the administration of Lu-177 PSMA-617 prior to treatment with docetaxel results in superior PSA response rates at 12 months as compared to the treatment with docetaxel alone. This study could help define the optimal sequencing of RLT with chemotherapy in hormone-sensitive diseases [107].

In taxane-naïve patients with mCRPC who have experienced disease progression on a second-generation ARPI (e.g., abiraterone, enzalutamide, darolutamide, or apalutamide), the PSMAfore (NCT04689828) phase III trial is being conducted to compare the efficacy of Lu-177 PSMA-617 against switching to an alternative ARPI. The study sought to determine whether RLT offers superior disease control and PFS over the standard practice of switching androgen receptor-targeted agents [138].

The phase I/II LuTectomy (NCT04430192) trial seeks to evaluate the dosimetry, safety, and efficacy of Lu-177 PSMA-617 in patients with high-risk localized or locally advanced PCa undergoing radical prostatectomy and pelvic lymph node dissection. By assessing its neoadjuvant potential, the trial aims to determine whether preoperative RLT can reduce the tumor burden and improve surgical outcomes [139].

The NEPI trial (EudraCT 2021-004894-30) investigates neoadjuvant Lu-177 PSMA-617, with or without the immune checkpoint inhibitor ipilimumab, in very high-risk PCa patients. The trial aims to assess the potential immunomodulatory effects of RLT and its role in enhancing long-term oncologic control [140].

Beyond PCa, RLT is also being evaluated as a first- or second-line treatment option in patients with gastroenteropancreatic neuroendocrine tumors (GEP-NETs). The phase III NETTER-2 (NCT03972488) clinical trial assesses whether the combination of Lu-177 DOTATATE (LUTATHERA) with long-acting octreotide improves the PFS in patients with high-proliferation GEP-NETs in comparison with high-dose (60 mg) octreotide monotherapy. The study seeks to determine whether RLT should be incorporated earlier into the management of aggressive neuroendocrine tumors [141]. Additionally, the COMPOSE (NCT04919226) trial evaluated the role of Lu-177 edotreotide peptide receptor radionuclide therapy (PRRT) as a first- or second-line treatment in patients with well-differentiated grade 2 and 3 somatostatin receptor-positive GEP-NETs. The study compares PRRT against the current SOC, with the aim of establishing whether the early integration of RLT can provide superior disease control as compared with conventional treatment options [142].

The shift toward earlier implementation of RLT represents a paradigm change in the management of metastatic PCa and neuroendocrine tumors. Although promising, the use of RLT in earlier disease settings must be carefully evaluated in terms of patient selection, treatment sequencing, and long-term safety. The results of these ongoing trials will be critical in shaping future clinical guidelines and determining whether RLT can be effectively integrated into standard oncologic care for patients at different disease stages.

## 7. Limitations and Considerations in the Clinical Field of Nuclear Medicine

### 7.1. Reimbursement and Economic Considerations

Establishing clear and comprehensive reimbursement pathways is an essential cornerstone for the successful implementation and sustainability of any new therapy, particularly in the complex field of radiotheranostics. Reimbursement frameworks must go beyond simply covering the cost of the therapeutic drug itself; they must also account for the numerous medical, technical, and procedural components integral to delivering effective care. These include imaging costs, dosimetry calculations, equipment use, and specialized professional services provided by nuclear medicine physicians, radiopharmacists, and other experts [143]. Radiotheranostic treatments are often administered by specialists who are not the patient’s primary physician, creating a need for seamless coordination among stakeholders, including primary care providers, referring specialists, and treatment centers. Aligning the incentives of all parties involved—healthcare providers, insurers, and patients—is crucial to fostering cooperation and ensuring access to these therapies.

Even highly effective and clinically valuable treatments can fail to gain traction if the reimbursement systems are not adequately developed. A lack of clarity or consistency in the reimbursement policies can discourage healthcare providers from offering these treatments, ultimately limiting patient access. Currently, the reimbursement landscape for radiotheranostic therapies is poorly defined and considerably varies across countries. For instance, while some progress has been made in securing reimbursements for groundbreaking treatments such as Lu-177 PSMA for PCa, these successes are exceptions rather than the normal consensus. In many cases, the disparities in coverage and funding create barriers to the widespread adoption of these innovative therapies and limit their availability [143].

To address these challenges, substantial investments in research and development are required, particularly for conducting well-designed, interdisciplinary, prospective clinical trials. These trials not only provide critical evidence of the clinical value and cost-effectiveness of radiotheranostics but also serve as a foundation for engaging policymakers, insurers, and other stakeholders in developing equitable reimbursement models. Collaboration among healthcare providers, industry leaders, and regulatory bodies will be key to creating standardized frameworks that ensure adequate compensation for all aspects of care delivery. Moreover, establishing global standards for coding, billing, and reimbursement processes will be essential for reducing variability and fostering consistency across the healthcare systems. Ultimately, a well-structured reimbursement system will not only promote the widespread adoption of radiotheranostics but also support its long-term sustainability and growth.

The advancement of radiotheranostics and the approval of new therapies are driving an increased need for specialized infrastructure to effectively evaluate and treat patients. This growth comes with several multifaceted challenges. A primary concern is the technical complexity of establishing fully operational radiotheranostic centers. Access to essential radioactive materials is highly regulated, and supply constraints arise from aging nuclear reactors, insufficient investment in modern production facilities, and inconsistencies with good manufacturing practices [101]. Additionally, the disparities in global access to radioisotopes pose a considerable barrier, emphasizing the necessity for coordinated efforts to expand production and improve distribution networks to meet growing demands.

### 7.2. Providing a Suitable Clinical Workforce and Environment

Another pressing issue is the shortage of skilled professionals, including nuclear medicine physicians, nuclear physicists, and radiopharmacists, who are important for the successful operation of these centers. Addressing this workforce gap requires robust training programs to enhance the expertise of current professionals and initiatives to attract and develop new talents in the field [144]. Moreover, standardized operational protocols and an emphasis on interdisciplinary collaboration are essential to ensure seamless patient care. Establishing efficient referral pathways among oncologists, urologists, gynecologists, and nuclear medicine specialists will facilitate timely patient access to theranostic treatments.

The key factors for the successful implementation of radiotheranostic strategies include integrating nuclear medicine experts into multidisciplinary tumor boards, clearly defining the patient care pathways to maintain continuity of care with the referring physicians, ensuring easy access to theranostic procedures, and addressing the complexities of coding and reimbursement. Additionally, fostering public and private investments in infrastructure and research, improving international cooperation for isotope production, and advancing technology to streamline processes will be critical for scaling up radiotheranostic capabilities globally.

### 7.3. Management of Salivary Gland Toxicity and Emerging Radiotracers in PSMA-Targeted Theranostics

The accumulation of PSMA-targeting radioligands in the major salivary glands presents a considerable challenge in the application of targeted radiotherapy for PCa [145]. This off-target uptake, comparable in magnitude to ligand sequestration within the prostate, frequently results in irreversible glandular damage, leading to xerostomia and a substantial decline in the quality of life of the affected patients [146]. Initial efforts to mitigate sialotoxicity, including external cooling with ice packs and sialendoscopy with steroid injections, have demonstrated limited efficacy, suggesting that inflammation alone is not the primary driver of radiotracer accumulation [147,148]. More promising results have been achieved with the intraparenchymal administration of botulinum toxin, which has been shown to considerably reduce the PSMA ligand uptake in the treated parotid glands [149]. However, this approach disrupts the neural control of salivation and is limited to the directly injected glands, potentially leading to extended glandular dysfunction [150].

Given the limitations of current strategies, research has shifted toward understanding and modulating the mechanisms of PSMA ligand uptake in the salivary glands. Given that most PSMA ligands are small molecule anions containing glutamate moieties, the potential for displacement using exogenous glutamate has been investigated in multiple studies [151,152]. Preclinical studies have demonstrated that intraperitoneal administration of monosodium glutamate (MSG) reduces PSMA ligand retention in the salivary glands and kidneys without considerably affecting tumor uptake [153]. However, the observed reductions may be more attributed to hypernatremia-induced physiological changes, including altered salivary gland perfusion, rather than direct ligand displacement. Current clinical trials are evaluating the co-administration of MSG to mitigate off-target salivary gland uptake in PSMA PET imaging [153,154]. Additionally, pilot studies have explored the use of cold DCFPyL instillation into the salivary glands as a potential strategy to prevent xerostomia in patients undergoing PSMA-targeted RLT [145]. Despite these advances, long-term salivary gland toxicity remains a critical concern, necessitating further investigation into effective and durable protective measures.

Concurrently, the emergence of novel radionuclides, including copper-based PET tracers, is poised to enhance the theranostic landscape. Cu-64 exhibits favorable imaging properties due to its positron emission and high-resolution PET capabilities, whereas its beta and Auger electron emissions offer potential for targeted radiotherapy [155]. The long half-life of Cu-64 allows for centralized production and widespread distribution, making it a practical alternative for clinical application. In addition, copper plays a vital role in biological processes, including cellular respiration and redox reactions, with elevated serum copper levels observed in various malignancies, correlating with the disease stage. Initial studies have demonstrated the feasibility of 64CuCl2 PET/CT for PCa staging, with detection rates comparable to those of multiparametric MRI and superior metastatic detection compared to F-18 Choline PET/CT. Importantly, 64CuCl2 PET/CT exhibits minimal urinary excretion, reducing interference with pelvic imaging, and no organ-specific toxicity has been reported despite the high hepatic uptake [156]. Although the theranostic potential of 64CuCl2 in PCa has been primarily investigated in preclinical models, ongoing studies are evaluating its role in targeted radiotherapy.

Given that PSMA-targeted RLT continues to evolve, optimizing treatment efficacy while minimizing off-target toxicity remains a key priority. The development of protective strategies against salivary gland toxicity, coupled with the introduction of novel radiotracers, such as Cu-64, represents a considerable advancement in precision oncology and personalized treatment approaches for PCa.

## 8. Conclusions

Nuclear medicine is increasingly recognized as a pivotal component in the diagnostic and therapeutic management of urological malignancies. The remarkable advancements in SPECT/CT and, more notably, PET/CT have enabled more precise tumor assessment across staging, recurrence, and treatment response scenarios. Unlike conventional anatomical imaging, molecular imaging provides insights into the biological and molecular alterations within tumors, allowing for the detection of disease processes that may not be evident when using standard imaging modalities. Consequently, PET/CT imaging is becoming an indispensable tool in specific clinical contexts, offering well-defined and tailored indications according to the urological neoplasm type and behavior.

A particularly dynamic area of innovation within nuclear medicine is theranostic approaches, which have made considerable strides in PCa treatment. This approach, which seamlessly integrates diagnostic and therapeutic capabilities into a single platform, has evolved into a well-established and routine treatment modality, particularly for mCRPC. PSMA radioligands, in particular, have revolutionized both the clinical workup and treatment strategies for mCRPC patients. These radioligands facilitate not only precise imaging and staging of metastatic disease but also targeted RLT, offering a highly specific and effective treatment option for patients with advanced disease.

Looking forward, the clinical utility of PSMA-based technologies is expected to considerably broaden. Ongoing research aims to refine the role of PSMA PET imaging in initial staging, as well as in monitoring the therapeutic response and guiding clinical decisions. Moreover, efforts are underway to optimize and standardize the use of PSMA-targeted RLT, potentially improving clinical outcomes further. Given the high global incidence of PCa, the anticipated increase in demand for RLT presents both considerable logistical challenges and promising opportunities for nuclear medicine departments worldwide.

Theranostic nuclear medicine embodies the principles of personalized medicine by delivering tailored therapeutic strategies based on an individual’s tumor biology. The unique concept of employing the same molecular construct for both diagnostic imaging and therapeutic interventions has paved the way for more targeted and effective treatments, particularly for mCRPC patients. This approach has been associated not only with prolonged survival but also with the preservation of patients’ quality of life, as it is generally well tolerated with low rates of adverse effects. As research progresses, there is growing optimism that theranostic approaches may be extended to earlier stages of PCa, potentially offering curative benefits and altering the disease course for a broader patient population.

## Figures and Tables

**Figure 1 biomedicines-13-01132-f001:**
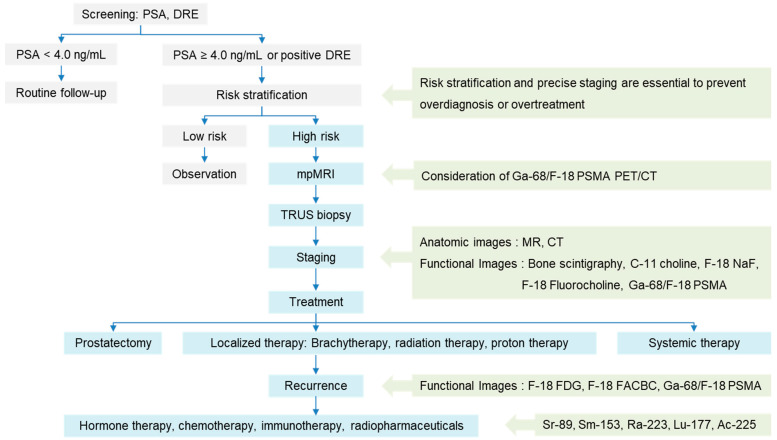
Flow diagram of prostate cancer diagnosis and treatment. PSA, prostate-specific antigen; TRUS, transrectal ultrasonography; MR, magnetic resonance; CT, computed tomography; PSMA, prostate-specific membrane antigen; FDG, fluoro-2-deoxyglucose; FACBC, trans-1-amino-3-F-18-fluorocyclobutanecarboxylic acid; Sr, strontium; Sm, samarium; Ra, radium; Lu, lutetium; Ac, actinium.

**Figure 2 biomedicines-13-01132-f002:**
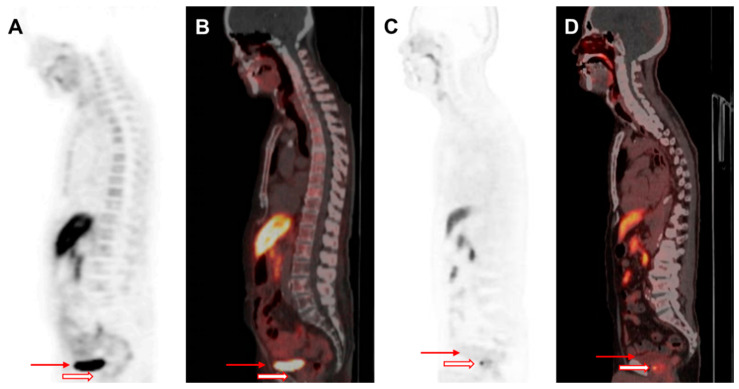
Arrows indicate bladder activity, whereas empty arrows denote the prostate gland. (**A**,**B**) represent the FCH PET/CT images of a 74-year-old man with mCRPC who underwent docetaxel treatment, demonstrating an intense urinary tracer activity in the bladder. (**C**,**D**) correspond to the F-18 PSMA PET/CT images of a 70-year-old man diagnosed with PCa (Gleason score of 9). (**C**) F-18 PSMA PET and (**D**) fusion images reveal minimal bladder activity and focal radiotracer uptake within the prostate, corresponding to the areas of PSMA overexpression. Reduced bladder activity may enhance the evaluation of primary prostate tumors.

**Figure 3 biomedicines-13-01132-f003:**
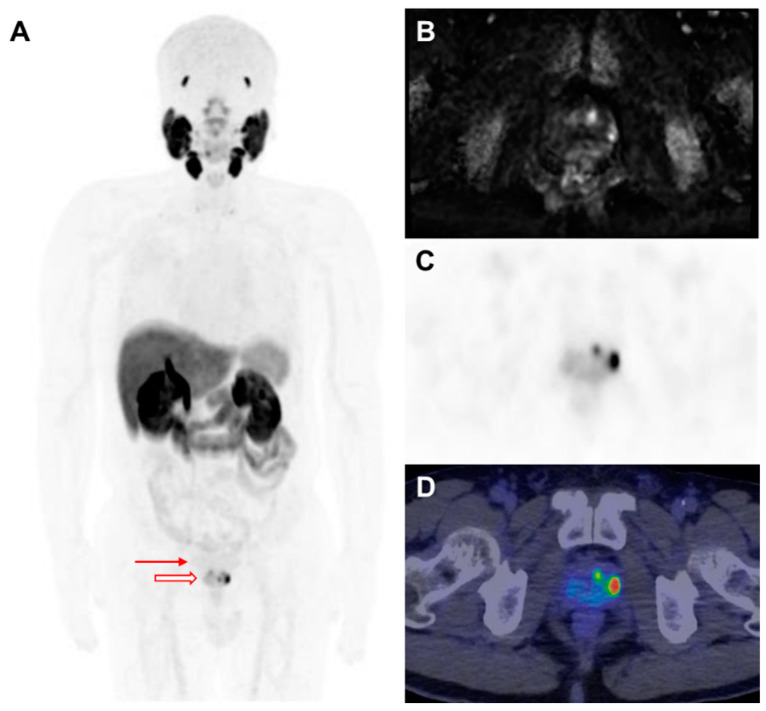
A 70-year-old man with a history of benign prostatic hyperplasia was referred to the urology department for lower urinary tract symptoms and had an initial serum prostate-specific antigen (PSA) level of 3.4 ng/mL. A transrectal ultrasound (TRUS)-guided biopsy confirmed adenocarcinoma with a Gleason score of 5 + 5 = 10. Immunohistochemical analysis showed a loss of high-molecular-weight cytokeratin (HMW CK) and p63, with positive expression of alpha-methylacyl-CoA racemase (AMACR). (**A**) Maximum intensity projection images of F-18 PSMA PET demonstrated a strong uptake in the primary tumor without distant metastases. A low bladder activity (arrow) improved the visualization of the pelvic tumor (empty arrow). (**B**) T2-weighted magnetic resonance imaging revealed hyperintense areas in the left superior and lateral prostate, corresponding to the TRUS biopsy sites. (**C**) F-18 PSMA PET and (**D**) PET/CT fusion imaging confirmed focal PSMA uptake in these regions, aligning with the MRI findings and biopsy-confirmed tumor locations.

**Figure 4 biomedicines-13-01132-f004:**
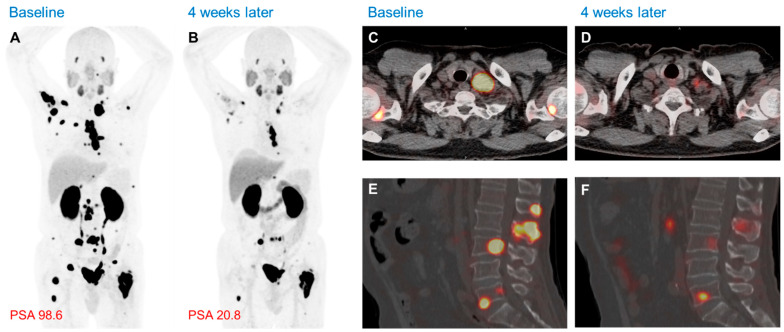
Clinical response to Lu-177 Ludotadipep (FC705; second generation of PSMA inhibitor labeled with Lu-177) by F-18 PSMA PET/CT in mCRPC [71]. (**A**) MIP image demonstrating multifocal radioactive uptake in the metastatic lymph nodes and bones. (**B**) After 4 weeks of treatment with Lu-177 Ludotadipep, the MIP image revealed a decrease in radioactive uptake as well as tumor burden in the previously noted multifocal radioactive uptake lesions. The PSA level also decreased from 98.6 ng/mL to 20.8 ng/mL, which is a suitable decline according to the PSA response. (**C**,**D**) are axial fusion images to show marked regression of the left supraclavicular lymph node and bone metastases. (**E**,**F**) are sagittal fusion images to note the decreased radioactive uptake in the lumbar spine lesions.

**Table 1 biomedicines-13-01132-t001:** Available radiotracers for prostate cancer imaging.

	Compound	Modality	Excretion	Mechanism of Action	Clinical Use
Skeletal imaging	Tc-99m MDP	SPECT	Renal	Chemisorption as a phosphate analog reflecting increased bone turnover	Initial evaluation in high risk ^1^ patients for skeletal metastases Elevated PSA, positive DRE, or symptomatic cases in postprostatectomy or definitive RT Regular response evaluation after ADT
	C-11 choline	PET	Hepatorenal	Lipid metabolism: cellular uptake and incorporation into the cell membrane and lipid synthesis by enhanced kinase expression in malignant PCa cells	Considered for equivocal lesions on initial bone scans Limited clinical use because of short half-life (20 min), requiring on-site cyclotron
	F-18 NaF	PET	Renal	Calcium analog: chemisorption of fluoride ions to the bone matrix by osteoblasts	High specificity, sensitivity, and PPV Alternative for conventional bone scans with better sensitivity, lesser uptake time, and higher resolution via hybrid images with CT
Soft tissue imaging	F-18 FDG	PET	Renal	Glucose metabolism: glucose analog surrogates for glycolysis (Warburg effect) Accumulation in malignant cells with enhanced glycolysis	High sensitivity but low specificity Not recommended for routine staging Prognostic implications in progressive CRPC
	F-18 fluorocholine	PET	Renal	Enhanced choline kinase activity in malignant PCa cells, thereby increasing choline transportation and phosphorylation	Role in biochemical recurrence and distant metastases
	F-18 fluciclovine (FACBC)	PET	Renal	Amino acid transport: amino acid leucine analog Cellular uptake by amino acid transporters and utilization by PCa cells	No significant renal excretion and short uptake time (4–10 min) Detection of biochemical recurrence and progression in extraprostatic disease Limited use of bone metastases due to significant marrow uptake
	Ga-68/F-18 PSMA	PET	Hepatorenal	Transmembrane glycoprotein-produced prostate epithelial cell membrane with PCa upregulation	High sensitivity and specificity for initial staging, biochemical recurrence, and progression in bone and soft tissues Good results with PSA level of <0.2 ng/mL ^1^ Need attention for “PSMA flare response” after ADT
	Tc-99m PSMA	SPECT	Renal	Transmembrane glycoprotein-produced prostate epithelial cell membrane with PCa upregulation	High sensitivity and specificity Easily accessed, simplicity, lower cost, long half-life

^1^ PSA of >20 ng/mL; more than T3 in the clinical stage; or with grade 4 or 5 disease (Gleason score of >7) [5]. RT, radiation therapy; ADT, androgen deprivation therapy; PPV, positive predictive value; FDG, fluoro-2-deoxyglucose; CRPC, castration-resistant prostate cancer; PSMA, prostate-specific membrane antigen; PSA, prostate-specific antigen; MDP, methylene diphosphonate.

**Table 2 biomedicines-13-01132-t002:** PSMA-targeting imaging agents and clinical applications.

Targeting Agents		Half-Life	Modality	Excretion	Clinical Application
Antibodies	In-111 7E11-C35 (capromab pendetide; ProstaScint)	67.2 h	SPECT	Renal	
	Zr-89 huJ591	78.4 h	PET	Renal	
	Zr-89 Df-IAB2M	78.4 h	PET	Renal	
Small-Molecule PSMA Ligands	Ga-68 PSMA-HBED-CC Ga-68 PSMA-11	68 min	PET	Renal	Initial staging and biochemical recurrence/progression Using therapy (Ga-68/Lu-177 DOTAGA-ffk(Sub-KuE)) Approved ^1^
	Ga-68 Ga-PSMA-I&T	68 min	PET	Renal	Initial staging and biochemical recurrence/progression
	F-18 DCFPyL (Pylarify^®^)	110 min	PET	Renal	Metastasis; approved ^1^
	F-18 PSMA-1007	110 min	PET	Hepatobiliary	Under phase III investigation
	F-18 Florastamin (FC-303)	110 min	PET	Renal	Under phase III investigation
	F-18 F-rhPSMA-7.3 (Posluma^®^)	110 min	PET	Renal	Approved ^1^
	Tc-99m PSMA ^2^	6 h	SPECT	Renal	Under phase III investigation

^1^ Refers to the regulatory approval for clinical use and distribution on a national or international level. ^2^ Includes Tc-99m MIP-1404, Tc-99m mas3-ynal-k(Sub-KuE), Tc-99m PSMA-T4, Tc-99m PSMA-11, Tc-99m HYNIC-PSMA, Tc-99m HYNIC-PSMA, and Tc-99m PSMA-I&S. PSMA, prostate-specific membrane antigen; In, indium; Zr, zirconium.

**Table 3 biomedicines-13-01132-t003:** Therapeutic radionuclides for prostate cancer.

Radionuclide	Half-Life	Projection	Target	Clinical Use
Imaging and therapeutic isotopes				
Lu-177	6.65 days	Reactor; β therapy	PSMA	mCRPC with progression on ADT and target-based chemotherapy
Sm-153	46.28 h	Reactor; β therapy	Hydroxyapatite of bone formation	Palliative purpose for symptomatic bone metastases
Therapeutic isotope				
Ra-223	11.43 days	Generator; α therapy	Osteomimic similar to calcium, leading to the combination of osteoblasts	mCRPC with symptomatic bone metastasis without visceral metastases
Ac-225	9.92 days	Reactor, cyclotron, generator; α therapy	PSMA	mCRPC

Lu, lutetium; PSMA, prostate-specific membrane antigen; mCRPC, metastatic castrate-resistant prostate cancer; ADT, androgen deprivation therapy; Sm, samarium; Ra, radium; Ac, actinium.

**Table 4 biomedicines-13-01132-t004:** Clinical trials of radioligand therapy in prostate cancer.

Study Title	Clinical Phase	Identifier	Radiopharmaceuticals	Objective	Conditions	Primary Endpoints
TheraP: A trial of Lu-177 PSMA-617 theranostic vs. cabazitaxel in progressive mCRPC (ANZUP Protocol 1603)	Phase II	NCT03392428	Lu-177 PSMA-617	Activity and safety evaluation of Lu-PSMA against cabazitaxel in progressive mCRPC	CRPC patients eligible for cabazitaxel as the next treatment Both PSMA PET/CT and FDG PET/CT: only for FDG-positive but PSMA-negative lesions Randomization of the two groups: LuPSMA radionuclide therapy against cabazitaxel	Reduction in PSA by at least 50% from baseline
VISION: Study of Lu-177 PSMA-617 In mCRPC	Phase III	NCT03511664	Lu-177 PSMA-617	Comparison of the two alternate primary endpoints of radiographic progression-rPFS and OS of the Lu-177 PSMA-617 therapy group and Lu-177 PSMA-617 with BSC/BSoC vs. BSC/BSoC alone group	mCRPC progression even after both ADT and either one or two taxane regimens PSMA-positive disease on Ga-68 PSMA-11 PET/CT Randomized 2:1 to receive Lu-177 PSMA-617 with/without BSC/BSoC	Imaging-based PFS and OS Improved PFS and OS in the Lu-177 PSMA-617 group
ALSYMPCA: Study of radium-223 dichloride in patients with symptomatic HRPC and skeletal metastases	Phase III	NCT00699751	Ra-223 dichloride (Xofigo, BAY88-8223)	Efficacy and safety of Ra-223 dichloride in patients with HRPC and skeletal metastases	Patients with symptomatic bone metastases	OS

ANZUP, Australian and New Zealand Urogenital and Prostate Cancer Trials Group; PSMA, prostate-specific membrane antigen; mCRPC, metastatic castrate-resistant prostate cancer; PET/CT, positron emission tomography–computed tomography; FDG, fluoro-2-deoxyglucose; LuPSMA, Lutetium-177-PSMA-617; PSA, prostate-specific antigen; rPFS, radiographic progression-free survival; OS, overall survival; BSC/BSoC, best supportive/best standard of care; ALSYMPCA, ALpharadin in SYMPtomatic Prostate Cancer; Ra, radium; HRPC, hormone-refractory prostate cancer.

**Table 5 biomedicines-13-01132-t005:** Clinical trials of combination therapy using radiopharmaceuticals in prostate cancer.

Study Title	Clinical Phase	Identifier	Radiopharmaceuticals	Objective	Conditions	Primary Endpoints
**ENZA-p:** Enzalutamide with Lu-177 PSMA-617 versus enzalutamide alone in men with mCRPC (ANZUP 1901)	Phase II	NCT04419402	Lu-177 PSMA-617	Activity and safety of adding Lu-177 PSMA to enzalutamide in mCRPC patients not previously treated with chemotherapy	mCRPC patients not previously treated with chemotherapy Randomized 1:1 to enzalutamide or enzalutamide and Lu-PSMA	PSA PFS
**LuPARP:** Lu-177 PSMA-617 therapy and olaparib in patients with mCRPC	Phase I	NCT03874884	Lu-177 PSMA-617	Safety and tolerability of olaparib in combination with Lu-177 PSMA in mCRPC	mCRPC progression on a novel AR-targeted agent and no prior experience with platinum, PARP inhibitors, or 177Lu-PSMA	DLTs and MTD
**LuPIN:** Lu-177 PSMA-617 and idronoxil in men with end-stage mCRPC	Phase I/II	ACTRN12618001073291	Lu-177 PSMA-617	Safety and efficacy investigation combining Lu-177 PSMA-617 with idronoxil (NOX66), a radiosensitizer	Progressive mCRPC previously treated with taxane chemotherapy and novel ASI	PSA response defined as 50% or PSA PFS/OS
**PRINCE:** Radionuclide Lu-177 PSMA-617 therapy in combination with pembrolizumab for the treatment of mCRPC	Phase Ib/II	NCT03658447	Lu-177 PSMA-617	Safety, tolerability, and efficacy of the combination of Lu-177 PSMA-617 and pembrolizumab in mCRPC	Minimal symptoms and progression on at least one line of novel AR-targeted agents	PSA response defined as ≥50% decrease from baseline
**UpFrontPSMA:** Randomized study of sequential Lu-177 PSMA 617 and docetaxel vs. docetaxel in mHNPC	Phase II	NCT04343885	Lu-177 PSMA-617	Effectiveness of Lu-177 PSMA with docetaxel against treatment with docetaxel only in newly diagnosed high-volume mCRPC patients	Newly diagnosed mCRPC Randomized 2:1 to Lu-177 PSMA and docetaxel against treatment with docetaxel only	Undetectable PSA (≤0.2 ng/mL) at 12 months after the commencement of protocol therapy
**AlphaBet:** Combination of Ra-223 and Lu-177 PSMA-I&T in men with mCRPC	Phase I/II	NCT05383079	Lu-177 PSMA-I&T	Safety of Ra-223 with Lu-177 PSMA-I&T in mCRPC	mCRPC patients who progressed on the second-generation AR antagonist	50% PSA response rate
**DORA:** Docetaxel vs. docetaxel and Ra-223 for mCRPC	Phase III	NCT03574571	Ra-223	Effectiveness of Ra-223 with docetaxel against treatment with docetaxel only in mCRPC patients	Documented progressive mCRPC cases with two or more bone metastases	OS defined as the time from randomization to death from any cause within 2 years

PSMA, prostate-specific membrane antigen; mCRPC, metastatic castrate-resistant prostate cancer; PSA, prostate-specific antigen; DLTs, dose-limiting toxicities; MTD, maximum tolerated dose; ASI, androgen signaling inhibitor; PFS, progression-free survival; OS, overall survival; PRINCE, PSMA-lutetium Radionuclide Therapy and ImmuNotherapy in Prostate CancEr; AR, androgen receptor.

## Data Availability

Data sharing is not applicable.

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
