# Peer review of "Current Status and Future Perspectives of Nuclear Medicine in Prostate Cancer from Imaging to Therapy: A Comprehensive Review"

_biomedicines, 2025, doi:10.3390/biomedicines13051132_

Round 1
Reviewer 1 Report
Comments and Suggestions for Authors
biomedicines-3583289 fully reviews the currently used imaging techniques in clinics for Prostate cancer's diagnosis and therapeutics, including advantages, limitations, and cautions upon separate or conjugated use among those imaging techniques, also provides an insightful information regarding how to select the radiotracers to enhance sensitivity and specificity for the diagnosis and therapeutic efficacy. Due mainly to the incidence and death rate of this cancer in male patients, this comprehensive review has a tremendous significance and interests in the field. The writing is good with appropriate cited references. It can be published as it except for minor modification of some cited references for their format or incomplete journal information, such as #5, #19, #59, (bold type for the year), #13 (lacking journal info), #124, #130 (missing page #).
Author Response
Biomedicines-3583289 fully reviews the currently used imaging techniques in clinics for Prostate cancer's diagnosis and therapeutics, including advantages, limitations, and cautions upon separate or conjugated use among those imaging techniques, also provides an insightful information regarding how to select the radiotracers to enhance sensitivity and specificity for the diagnosis and therapeutic efficacy. Due mainly to the incidence and death rate of this cancer in male patients, this comprehensive review has a tremendous significance and interests in the field. The writing is good with appropriate cited references. It can be published as it except for minor modification of some cited references for their format or incomplete journal information, such as #5, #19, #59, (bold type for the year), #13 (lacking journal info), #124, #130 (missing page #).
REPLY: We are very much thankful to the reviewer for the thorough review. We agree with all the specific comments raised and have revised our paper in light of the useful suggestions.
We appreciate the reviewer for taking time to carefully review the manuscript and give detailed and constructive comments, which has greatly helped to improve the paper. After reviewing our manuscript, we apologize for our technical errors, and have modified our references as suggested. Concerning reference 13, as the reference being an internet book publish, there were limitations to citation formation. However, we have modified the reference as close to format as possible. We ask for the reviewer’s understanding for this shortcoming. As we have made corrections and have additional content to our manuscript, there has been an addition to our reference list and the numbers to our references have changed accordingly. To improve clarity, we have indicated modifications in red font in our revised manuscript. Thank you for your insight.

Reviewer 2 Report
Comments and Suggestions for Authors
This review paper is well-conceived and clearly structured, providing a comprehensive overview of the topic. The article is well-structured and thoughtfully developed, providing a thorough and current overview of the application of nuclear medicine in the diagnosis and treatment of urological malignancies, with a strong focus on prostate cancer. The abstract clearly outlines the major advancements in imaging technologies such as SPECT/CT and PET/CT, as well as the growing clinical significance of theranostics, particularly the use of PSMA-targeted radioligand therapy. The discussion demonstrates a solid understanding of both the clinical and research landscapes, effectively highlighting the benefits of personalized, precision oncology. The references are relevant, recent, and well-integrated throughout the text, supporting the scientific arguments presented. Additionally, the language is fluent and professionally written, which ensures clarity and facilitates reader engagement. Overall, the paper offers valuable insights into an evolving and highly impactful area of nuclear medicine, making it a significant contribution to the field.

Author Response
The manuscript titled "Current Status and Future Perspectives of Nuclear Medicine in Prostate Cancer from Imaging to Therapy: A Comprehensive Review" is a well-organized and expertly written article that provides a thorough and up-to-date overview of the role of nuclear medicine in prostate cancer management. It integrates the latest advancements in imaging and therapy, with a strong emphasis on theranostics and PSMA-targeted strategies. The review is not only scientifically rigorous but also clearly structured and highly informative, making it a valuable contribution to the field.
The introduction effectively presents the global impact and heterogeneity of prostate cancer, emphasizing the need for tailored therapeutic strategies. The authors clearly establish the relevance of nuclear medicine in this context and provide a compelling rationale for the review, setting the stage for a comprehensive discussion. In section Nuclear Medicine Approaches for Diagnosis and Staging of Prostate Cancer is presented a clear and insightful discussion on the evolution of diagnostic imaging, especially the shift from conventional anatomical imaging to functional and hybrid modalities like PET/CT and SPECT/CT. The integration of these tools into staging and treatment planning is well articulated and clinically relevant. A particularly strong section of the manuscript “Function of Radiotracers in Prostate Cancer Imaging “offers a detailed overview of available radiotracers, their mechanisms of action, and comparative performance. The inclusion of summary tables adds significant value, aiding in the practical understanding of their clinical utility. The focus on PSMA-targeted tracers reflects the most current and impactful trends in the field. Further, the authors provide a well-balanced examination of radioligand therapies (RLT), highlighting both their promise and limitations. The section is rich in data and clinical trial references, supporting the therapeutic potential of agents such as Lu-177, Ra-223, and Ac-225. This discussion underscores the paradigm shift toward precision oncology.
By summarizing key clinical trials such as TheraP and VISION, the authors demonstrate the growing evidence base supporting PSMA-targeted therapies. This section “Current Clinical Trials and Future Perspectives” is particularly useful for understanding how nuclear medicine is rapidly becoming an integral component of advanced prostate cancer care. This forward-looking section “Future Directions of Nuclear Medicine in the Clinical Setting of Prostate Cancer” enhances the manuscript’s value by exploring emerging areas such as novel radionuclides, improved chelation chemistry, combination therapies, and radioguided surgery. The authors show great awareness of the challenges and opportunities ahead, providing a thoughtful roadmap for future research and clinical application.
In conclusion, this is a high-quality manuscript that offers both comprehensive coverage and critical insight into the rapidly advancing field of nuclear medicine in prostate cancer.
The content is well-researched, current, and highly relevant for clinical and academic audiences. I fully support its publication.
REPLY: We sincerely thank the reviewer for their thorough and thoughtful evaluation of our manuscript titled "Current Status and Future Perspectives of Nuclear Medicine in Prostate Cancer from Imaging to Therapy: A Comprehensive Review." We are grateful for the positive feedback highlighting the organization, scientific rigor, clarity, and relevance of our work.
We particularly appreciate the reviewer’s recognition of our efforts to provide a comprehensive and current overview of nuclear medicine approaches in prostate cancer management, including the evolution of diagnostic imaging, the role of PSMA-targeted strategies, and the emerging applications of radioligand therapies. We are pleased that the inclusion of summary tables and the discussion of key clinical trials such as TheraP and VISION were found to be valuable and informative.
Our gratitude goes to the reviewer for acknowledging our focus on future directions, including novel radionuclides, combination therapies, and radioguided surgery, which we believe are critical areas for continued advancement in the field.
The authors are encouraged by the reviewer’s support for the publication of our manuscript and remain committed to contributing meaningful and clinically relevant insights to the evolving landscape of nuclear medicine in prostate cancer care.
Thank you once again for your constructive and encouraging feedback.

Reviewer 3 Report
Comments and Suggestions for Authors
In the article "Current Status and Future Perspectives of Nuclear Medicine in Prostate Cancer from Imaging to Therapy: A Comprehensive Review," the authors present the importance of nuclear medicine in the diagnosis and treatment of prostate cancer, with an emphasis on advanced PET/CT molecular imaging techniques and targeted radiotherapy. Overall, the article is well written and organized. Among the specific radiotracers reviewed were F-18 and Ga-68 PSMA, F-18 fluorocholine, and F-18 fluciclovine, which are useful in detecting biochemical recurrence and metastases. Radiotherapy modalities included beta therapy (Lu-177 PSMAs) and alpha therapy (Ra-223 chloride and Ac-225 PSMAs). Although the authors mentioned that PET/CT and SPECT molecular imaging modalities provide functional information with high diagnostic accuracy in prostate cancer, the review included only PET radiotracers. Currently, 99mTc-radiolabeled PSMA derivatives for SPECT imaging are numerous and have demonstrated their usefulness in the primary diagnosis of prostate cancer. Despite the lower lesion detection rate of 99mTc-PSMA compared to 18F-PSMA PET/CT, there is no impact on clinical staging and consequently on initial treatment intent. Therefore, I suggest that the authors include Tc-99m-PSMA radiopharmaceuticals in this review.
Author Response
In the article "Current Status and Future Perspectives of Nuclear Medicine in Prostate Cancer from Imaging to Therapy: A Comprehensive Review," the authors present the importance of nuclear medicine in the diagnosis and treatment of prostate cancer, with an emphasis on advanced PET/CT molecular imaging techniques and targeted radiotherapy. Overall, the article is well written and organized. Among the specific radiotracers reviewed were F-18 and Ga-68 PSMA, F-18 fluorocholine, and F-18 fluciclovine, which are useful in detecting biochemical recurrence and metastases. Radiotherapy modalities included beta therapy (Lu-177 PSMAs) and alpha therapy (Ra-223 chloride and Ac-225 PSMAs). Although the authors mentioned that PET/CT and SPECT molecular imaging modalities provide functional information with high diagnostic accuracy in prostate cancer, the review included only PET radiotracers. Currently, 99mTc-radiolabeled PSMA derivatives for SPECT imaging are numerous and have demonstrated their usefulness in the primary diagnosis of prostate cancer. Despite the lower lesion detection rate of 99mTc-PSMA compared to 18F-PSMA PET/CT, there is no impact on clinical staging and consequently on initial treatment intent. Therefore, I suggest that the authors include Tc-99m-PSMA radiopharmaceuticals in this review.
REPLY: We are very much thankful to the reviewer for the thorough review. We agree with all the specific comments raised and have revised our paper in light of the useful suggestions.
While PSMA PET/CT has emerged as a valuable imaging modality for prostate cancer, particularly in initial staging, detection of biochemical relapse, and guidance of targeted therapy, its broader clinical adoption remains constrained. These limitations are largely due to restricted access to PET infrastructure, the need for specialized radiation safety measures, and the relatively high cost of the modality—issues that are especially pronounced in remote or resource-limited settings. In contrast, SPECT imaging is more widely available across healthcare systems worldwide. Tc-99m-labeled radiotracers used in SPECT are inexpensive, easy to produce, and offer a longer half-life, making them well-suited for routine clinical use in nuclear medicine. Moreover, technological advancements in SPECT have significantly enhanced image resolution, narrowing the gap between SPECT and PET in terms of diagnostic quality and thereby supporting its increased clinical application. Importantly, while advanced imaging modalities capable of detecting additional metastatic lesions may improve diagnostic certainty, their added clinical value is often limited. In most cases, the detection of a greater number of metastatic sites does not substantially alter therapeutic decision-making for PCa patients with metastatic disease. The exception lies in scenarios where a given imaging modality fails to detect any metastases, potentially influencing decisions regarding eligibility for systemic treatments or radioligand therapy. Thus, the true clinical utility of an imaging modality should be evaluated based not only on its sensitivity but also on whether it provides actionable insights that directly inform and influence patient management strategies.
In concert with the reviewer’s suggestions, we have incorporated additional information onTc-99m-PSMA radiopharmaceuticals in section 3.2.6 “small molecule ligands” of our revised paper, and have revised Tables 1 and 2 for better reading fluency. Moreover, we have highlighted the revised material in red for improved clarity. The authors thank the reviewer for the thoughtful comments.
